# Scalable Production of Size-Controlled Cholangiocyte and Cholangiocarcinoma Organoids within Liver Extracellular Matrix-Containing Microcapsules

**DOI:** 10.3390/cells11223657

**Published:** 2022-11-18

**Authors:** Gilles S. van Tienderen, Jorke Willemse, Bas van Loo, Eline V. A. van Hengel, Jeroen de Jonge, Luc J. W. van der Laan, Jeroen Leijten, Monique M. A. Verstegen

**Affiliations:** 1Department of Surgery, Erasmus MC Transplant Institute, University Medical Center Rotterdam, 3015 GD Rotterdam, The Netherlands; 2Leijten Laboratory, Department of Developmental BioEngineering, Faculty of Science and Technology, University of Twente, 7522 NB Enschede, The Netherlands

**Keywords:** organoids, microcapsules, microfluidics, drug screening, liver tissue engineering, cholangiocarcinoma

## Abstract

Advances in biomaterials, particularly in combination with encapsulation strategies, have provided excellent opportunities to increase reproducibility and standardization for cell culture applications. Herein, hybrid microcapsules are produced in a flow-focusing microfluidic droplet generator combined with enzymatic outside-in crosslinking of dextran-tyramine, enriched with human liver extracellular matrix (ECM). The microcapsules provide a physiologically relevant microenvironment for the culture of intrahepatic cholangiocyte organoids (ICO) and patient-derived cholangiocarcinoma organoids (CCAO). Micro-encapsulation allowed for the scalable and size-standardized production of organoids with sustained proliferation for at least 21 days in vitro. Healthy ICO (*n* = 5) expressed cholangiocyte markers, including KRT7 and KRT19, similar to standard basement membrane extract cultures. The CCAO microcapsules (*n* = 3) showed retention of stem cell phenotype and expressed LGR5 and PROM1. Furthermore, ITGB1 was upregulated, indicative of increased cell adhesion to ECM in microcapsules. Encapsulated CCAO were amendable to drug screening assays, showing a dose-response response to the clinically relevant anti-cancer drugs gemcitabine and cisplatin. High-throughput drug testing identified both pan-effective drugs as well as patient-specific resistance patterns. The results described herein show the feasibility of this one-step encapsulation approach to create size-standardized organoids for scalable production. The liver extracellular matrix-containing microcapsules can provide a powerful platform to build mini healthy and tumor tissues for potential future transplantation or personalized medicine applications.

## 1. Introduction

Hepatobiliary organoids are valuable biomimetic ex vivo models for regenerative medicine and disease modeling purposes [1,2,3]. These three-dimensional organotypic cell structures recapitulate aspects of the native liver in vitro through complex cell–cell and cell–matrix interactions, which allow for self-organization [4,5]. Healthy donor-derived human cholangiocyte organoids used in a regenerative setting have shown the potential to restore damaged biliary epithelium in a preclinical model [6,7]. Simultaneously, patient-derived cholangiocyte and cholangiocarcinoma organoids offer an interesting platform for studying biliary diseases, particularly for drug screening applications [2,8,9,10]. However, the clinical and industrial translation of these organoids is currently limited by the use of mouse tumor-derived basement membrane extracts (BME).

Organoids are typically cultured in BME, which is a complex mixture of extracellular matrix (ECM) components derived from the mouse tumor mass produced by Englebreth-Holm-Swarm cells [11,12]. BME allows for self-organization of the cells into three-dimensional organotypic structures with maintenance of a proliferative status and can be used to create a tumor tissue-like microenvironment in vitro [5,13,14,15]. However, the process of cellular self-organization is uncontrolled, leading to relatively large inter-organoid disparity in size [16]. Organoid size is known to influence cell growth and division and thus introduces unwanted heterogeneity for standardization within translational applications of organoids [17,18]. This results in limited reproducibility of in vitro experiments. Furthermore, expansion of organoids in BME hemispheres is labor intensive and time consuming, which reduces scalability of the organoid cultures [19]. Thus, to overcome these limitations, there is a need for a controlled, continuous, high-throughput organoid production processes utilizing BME alternatives that are good clinical practice compliant [20].

We have recently shown that BME can be replaced by a hydrogel derived from decellularized human liver ECM [21]. This hydrogel supports the initiation and culture of cholangiocyte organoids and can unlock direct clinical applications for regenerative medicine purposes [13]. However, the use of this hydrogel does not resolve the high level of heterogeneity in organoid size and culture and still requires a time-consuming process. The incorporation of decellularized liver ECM with bioengineering encapsulation strategies could provide the necessary tools to comprehensively circumvent the persisting limitations.

In this study, we report on a microfluidic methodology for the controlled, high-throughput production of intrahepatic cholangiocyte organoids (ICO) and cholangiocarcinoma organoids (CCAO) in hybrid microcapsules containing decellularized liver-derived extracellular matrix [21]. In order to achieve this, primary human cholangiocyte and cholangiocarcinoma cells are encapsulated in a human liver extracellular matrix-containing dextran-tyramine microcapsules utilizing single-step enzymatic outside-in crosslinking in an off-the-shelf, reusable, flow-focusing microfluidic device. Encapsulated organoids are of similar quality compared to standard BME-cultured organoids but show a high standardization with regard to size. In addition, this setup allows for the production of high quantities of tumor organoids that are similar in size and quality for scalable personalized drug screening.

## 2. Materials and Methods

### 2.1. Human Donor Liver Biopsies and Cholangiocarcinoma Biopsies for Organoid Cultures

Intrahepatic cholangiocyte organoids (ICO) (*n* = 5) were initiated from donor liver biopsies (0.5 cm^3^–1 cm^3^) following a previously published protocol [5,22]. The use of liver biopsies for research purposes was approved by the medical ethics committee of the Erasmus University Medical Center (MEC-2014-060). Cholangiocarcinoma (CCA) tumor tissue biopsies (*n* = 3) were obtained through liver resections performed at the Erasmus MC (MEC-2013-143) and CCAO were initiated as described previously [9]. All patients gave written informed consent to use their tissue for research purposes. In short, the biopsies were digested in 2.5 mg/mL collagenase type A (Sigma-Aldrich, St. Louis, MO, US) for 20–120 min at 37 °C. Next, the suspension was strained (70 µm cell strainer), washed with cold ADV+ (Appendix A), resuspended in basement membrane extract (BME, Cultrex, R&D systems, Minneapolis, MN, United States), and plated in 25 µL droplets in 48-well suspension culture plates (Greiner Bio One, Alphen aan den Rijn, The Netherlands). The BME was allowed to solidify for 45–60 min at 37 °C before 250 µL startup expansion medium (SEM, Appendix A) was added. After 72 h, SEM was replaced with Expansion Medium (EM, Appendix A) and refreshed every 3 to 4 days. Organoids were passaged by mechanical dissociation every 7 days in 1:3 to 1:6 split ratios.

### 2.2. Human Liver for Decellularization and Hydrogel Preparation

Human livers (*n* = 4), deemed unsuitable for transplantation, were decellularized as previously described [23]. In short, livers were obtained after declination for liver transplantation by all transplant centers in the Eurotransplant zone. Informed consent for the use of these livers for research purposes was given to transplant coordinators of the Dutch Transplant Foundation by next of kin and was approved by the Erasmus MC medical ethics committee (MEC-2012-090).

Decellularization was achieved by continuous perfusion with Triton X-100 solution for 120 min, followed by nine 120 min reperfusion cycles with 10 L Tx100 solution. Afterward, the livers were flushed with 100 L dH2O and stored in dH2O for 10–14 days at 4 °C. This was followed by DNase treatment (10 mg/L DNase type I (Sigma) in 0.9% NaCl + 100 mM CaCl_2_ + 100 mM MgCl_2_) for eight hours at RT. Complete decellularization was confirmed based on histology and DNA quantification.

HLECM hydrogels were prepared as previously described [21]. In short, the ECM was lyophilized (Zirbus Technology Sublimator 400) and pulverized (Retsch ZM200). HLECM (40 mg/mL) was digested in 10% (*W/W*) Pepsin (Sigma, 3200–4500 U/mg) in 0.5 M Acetic Acid over a 72 h period at RT. Afterward, cold 10%(*V/V*) 10X PBS and cold 10% (*V/V*) ADV+ were added. PH was neutralized to 7.4–7.6 by the addition of 1 M NaOH. Solubilized HLECM concentration was measured using a Pierce BCA assay (ThermoFisher, Waltham, MA, United States) and, if needed, further diluted to 8 mg/mL. HLECM was stored at −20 °C.

### 2.3. Microfluidic Encapsulation

A reusable 3D microfluidic glass capillary platform was used for the production of droplets, as previously described [24]. A polymethylmethacrylate (PMMA) microfluidic device was fabricated using standard cutting and abrasion technology [24]. Fused silica capillaries (inner diameter of 200 µm; Polymicro Technologies, Phoenix, AZ, United States) were used to fabricate the nozzles by insertion into a transparent, semi-permeable silicone tubing (Helix Medical, Carpinteria, CA, United States) with a borosilicate capillary spacer (CM Scientific, Silsden, United Kingdom). The nozzles were operated in a flow-focusing configuration and enabled droplet formation, in-line monitoring of droplet formation, and allowed for diffusion-based delivery of H2O2 (Sigma-Aldrich) to controllably induce the enzymatic outside-in crosslinking of the formed droplets.

5% Dextran-tyramine (Dex-Ta) + 0.1% HLECM + 250 U/mL horseradish peroxidase (HRP) (Sigma-Aldrich) solutions were used as hydrogel precursor solution. The resulting Dex-Ta contained 14 tyramine moieties per 100 repetitive units. Droplets were produced in a N-Hexadecane (Sigma-Aldrich) oil phase supplemented with 1% Span 80 (Sigma-Aldrich) as a surfactant. Droplets were produced using a total flow rate of 100 µL/min with a water:oil ratio of 1:10 using a low-pressure syringe pump (neMESYS, Cetoni, Korbußen, Germany). The enzymatic outside-in crosslinking was induced through the transportation of droplets within a semi-permeable silicone tube that was immersed for 20 cm in a 30% H_2_O_2_ bath. Diffusion of H_2_O_2_ through the tubing and the oil phase into the Dex-Ta, HLECM, and HRP-containing droplets allowed for the formation of microcapsules. Produced microcapsules were collected in the oil. The emulsion was broken by five consecutive washes using N-Hexadecane and subsequent washes in phosphate-buffered saline (PBS).

For the production of organoid-laden microcapsules, cells were trypsinized, washed with medium, and collected using a 40 µm cell strainer. Cells were suspended in the hydrogel precursor solution at a concentration of 1 × 106 cells/mL, which was supplemented with 8% optiprep (Sigma-Aldrich) to obtain ρ = 1.05 g/L. The cell-laden hydrogel precursor solution was loaded in an ice-cooled gastight syringe (Hamilton) that was continually agitated using a small magnet in order to prevent the formation of cell clumps. Upon collection and washing, cell-laden microcapsules were divided into 12-well plates.

### 2.4. Cell Culture in Microcapsules

Encapsulated organoids were placed in EM, which was refreshed every 3–4 days. Cell viability was assessed using live dead staining. Organoids were incubated with propidium iodide (50 μg/mL) and calceinAM (0.5 μM) at 37 °C and 5% CO_2_ for 60 min. An EVOS microscope (ThermoFisher) was used to image cultures over time. The average diameter of ICO was measured using ImageJ (Version 1.53u, Bethesda, MD, USA).

At set time points (D7 and D21), organoids were fixed using 4% paraformaldehyde (Fresenius Kabi, Bad Homburg, Germany) or lysed in Qiazol (Qiagen, Hilden, Germany) lysis buffer. Lysed samples were stored at −80 °C.

### 2.5. FFPE Histology

Samples were washed in 1X PBS. The 1% (W/V) agarose was dissolved in PBS by heating the solution. Encapsulated organoids were placed in a 96-well plate, and PBS was removed. Agarose was added and allowed to solidify at 4 °C and subsequently embedded in paraffin. Paraffin blocks were sectioned at 4 µm thickness and stained with hematoxylin and eosin or used for immunohistological staining. For the latter, heat-induced epitope retrieval was performed using TRIS-EDTA buffer (pH = 8.0). Subsequently, slides were incubated in goat serum for 1 h before incubation of primary antibodies (listed in Appendix A) overnight at 4 °C. Slides were washed using PBS before incubation (1 h, RT) with secondary antibodies (listed in Appendix A). All slides were counterstained with DAPI (Vectashield anti-fade mounting medium with DAPI, Vectorlabs, Newark, CA, United States) and imaged on a Leica DM6000 CFS microscope with a LEICA TCS SP5 II confocal system. Data were processed and analyzed using ImageJ.

### 2.6. Whole Mount Confocal Microscopy

Fixated samples were permeabilized with 0.1% Triton X-100 in 1× PBS for 20 min. The 5% serum in 1× PBS was used to block samples for 60 min. Primary antibodies (Appendix A) were incubated overnight at 4 °C. Secondary antibodies (Appendix A) were incubated for 60 min. F-actin staining was performed by incubating samples with Alexa Fluor 488 Phalloidin (ThermoFisher) for 20 min at RT. All samples were counterstained with DAPI (Vectashield anti-fade mounting medium with DAPI, Vectorlabs). Samples were imaged using a Leica 20× water dipping lens on a Leica DM6000 CFS microscope with a LEICA TCS SP5 II confocal system. Data were processed and analyzed using ImageJ.

### 2.7. RT-qPCR

Qiazol samples were homogenized with a TissueRuptor (Qiagen, Hilden, Germany). RNA isolation was performed with the miRNeasy (Qiagen) kit according to the manufacturers’ protocol, and RNA was measured on the Nanodrop 2000. cDNA (2 ng/µL) was prepared using 5x PrimeScript RT Master Mix and a 2730 Thermal cycler (Applied Biosystems, Waltham, MA, USA). RT-qPCR was performed using SYBR select master mix for SFX (Applied Biosystems) on a StepOnePlus RT PCR system (Applied Biosystems). The primers used are listed in Appendix A.

### 2.8. In Vitro Drug Assay on Encapsulated CCAO

Preliminary to the drug response, encapsulated CCAO cell viability was measured using CellTiter-Glo (Promega, Madison, WI, USA) to equalize the relative amount of ATP in each organoid line to minimize drug response variability because of cell numbers. Next, encapsulated CCAO (*n* = 3) were plated out at a concentration of approximately 5000 cells/well in 96-well plates (Cellstar, Greiner Bio-One, Alphen aan den Rijn, The Netherlands). Encapsulated organoids were cultured for 24 h, and subsequently, a concentration dilution series of gemcitabine, range 0.01 µM–1000 µM (200 mg/5 mL, Sandoz, Basel, Switzerland), with a fixed concentration of cisplatin, 10 µM, (1 mg/mL, Accord) was added. Furthermore, a high-throughput drug screening using 51 drugs from the drug panel of approved oncology drugs (AOD X, NIH, dtp.cancer.gov, accessed on 15 May 2021) was added at a fixed concentration of 1 µM. After 72 h of incubation, cell viability was measured using CellTiter-Glo (Promega, Madison, WI, USA). An experimental concentration range was determined if possible, and dose-response curves were fitted using nonlinear least squares regression fitting.

### 2.9. Statistical Analysis

Data analysis was performed in Prism 8.0. Kruskal–Wallis test by ranks was performed on data sets with non-paired samples or different sample sizes. A Friedman test was performed on matched samples.

## 3. Results and Discussion

### 3.1. Healthy Intrahepatic Cholangiocyte Organoids (ICO) Can Self-Assemble in Microcapsules

ICO was produced by cholangiocyte cell encapsulation in microcapsules utilizing enzymatic outside-in crosslinking of cell-laden hydrogel precursor droplets as described by van Loo et al. [24] and as depicted in Figure 1A,B. Hydrogel precursor droplets flow through a semi-permeable silicone tubing, which allows for outside-in diffusion of hydrogen peroxide (H_2_O_2_), which initiates an outside-in crosslinking reaction that results in microcapsules (Figure 1C,D) with high monodispersity (coefficient of variation = 1.8%) (Figure 1C,D and Appendix A).

The main advantage of the outside-in enzymatic crosslinking approach is the simple single-step continuous production process it utilizes, which is in contrast to multistep approaches such as microcapsules utilizing sacrificial cores [25,26], complex multiple emulsion strategies [27,28], or non-continuous layer-by-layer biofabrication [29]. Most of these complex approaches are based on ionic crosslinking of alginate hydrogels, which is considered unstable due to the inherently reversible non-covalent crosslink that is susceptible to gradual loss of divalent ions from the crosslinked alginate [30]. The enzymatic covalent crosslinking of Dextran-TA results in stable microcapsules, which allow for long-term cell culture and microaggregation of microencapsulated cells [24].

Cells derived from healthy ICO were encapsulated in microcapsules of 5% Dextran-TA and 0.1% human liver extracellular matrix (HLECM) microcapsules. Dextran was chosen as it is a versatile and cytocompatible biomaterial that can be modified with relative ease (e.g., the addition of the functional tyramine group to the polymer) [31]. However, the addition of 0.1% HLECM was necessary as Dex-Ta hydrogel without HLECM did not support the self-organization of ICO (Appendix A). Alternative (synthetic) ECM components, such as collagen [14], cell adhesion motifs (fibronectin or RGD) [32], or laminin-111 [15,33], can be used to augment the inert polymeric backbone and to create a more tissue-mimetic or tumor surrounding [8,14,34]. These components were considered, but fine-tuning and optimizing the concentrations in designer hydrogels with ECM components can be cumbersome and challenging [13]. Alternatively, ECM extracts from decellularized tissues can be used [35]. Different publications have shown that organoids can be grown in hydrogels derived from different decellularized tissues, which include small intestinal submucosa [36] or stomach [37]. However, it was also shown that there is a tissue-specific effect of the ECM hydrogels on the organoids [37]. HLECM can successfully replace BME as a tissue-specific culture substrate for ICO; therefore, we tested this liver ECM extract to augment the Dex-Ta hydrogels [21].

The use of HLECM allowed for the formation of ICO inside Dex-Ta hydrogels and the Dex-Ta microcapsules. The cells formed aggregates after one day of culture, and these aggregates were cultured for up to 21 days inside the microcapsules (Figure 1E). The average diameter of encapsulated ICO did not significantly increase between day 7 (98 µm, SD: ±37 µm), day 14 (109 µm, SD: ±35 µm), or day 21 (112 µm, SD: ±36 µm). The average diameter of the ICO did not exceed the outer diameter (O.D.) of the microcapsules, which was 157 µm (SD: ±14 µm). Moreover, the difference between the average diameter for ICO grown in BME (332 µm, SD: ±259 µm) was significant (*p* < 0.001) when compared to encapsulated ICO. This indicates that encapsulated ICO are more size-controlled than ICO grown in BME, as there is less variation in organoid size (Figure 1F). ICO cells remained viable during the 21-day culture period (Figure 1G). Dead cells were observed inside the microcapsules; however, it remains unclear whether this is the result of the encapsulation process or normal cell death during the culture period. Similar levels of cell death are also seen in ICO grown in BME. However, in BME-grown ICO, dead cells typically accumulate in the lumen of the organoids, whereas with the encapsulated organoids, dead cells appear to accumulate in between the organoid and the capsule. Hematoxylin and eosin staining on FFPE sections showed that the encapsulated ICO has a lumen that is similar to the control ICO grown in BME (Figure 1H and Appendix A). Of note, the thickness of the cell layers appeared thicker in the encapsulated ICO, which could be due to the columnar polarization of the cholangiocytes inside the microcapsules. Similar phenotypes were seen for cholangiocyte organoids grown on ductal ECM when cell growth was limited by space (e.g., confluent cell layers) [38,39].

Encapsulated ICO stained positive for the cholangiocyte marker KRT7 (Figure 2A–C). Proliferation marker KI67 was found to be present at day 7 but was decreased at day 21 (Figure 2D), which might be due to the organoid size restriction due to the set microcapsule size. The zonula occludens 1 (ZO1) honeycomb structure was found both on days 7 and 21 (Figure 2E–G), indicating the formation of tight junctions at the interface of the lumen. Interestingly, in vivo ZO1 is only present inside the lumen of the bile duct, while the encapsulated ICO revealed the ZO1 honeycomb structure on both the inside and the outside of the organoids. This could indicate multiple polarized cell layers sandwiched together, with the outermost layer turned inside out. On a transcriptome level, encapsulated ICO maintained a similar expression profile as BME-cultured ICO for cholangiocyte markers KRT7, KRT19, and EPCAM, MUC1 (Figure 2H). This was also seen for stem cell marker LGR5, which in vivo is only upregulated in damaged liver tissue [40], but can be regarded as a marker for the cholangiocyte organoids. Hepatocyte-like cell marker albumin did not significantly change over the culture period. The proliferation marker KI67 appeared to decrease on the protein level by day 21 (Figure 2D) but not on the mRNA level (Figure 2H). Actually, an increasing trend of KI67 gene expression was observed at 21-day, though this did not reach statistical significance. Average gene expression of epithelial-mesenchymal transition marker vimentin remained stable, indicating that ICO does not undergo this transition after encapsulation. In conclusion, encapsulated ICO resemble ICO grown in BME on gene- and protein expression level for selected markers.

Human cholangiocyte organoids are of interest for regenerative applications, such as ex vivo repair of the damaged biliary epithelium [6,13]. Although not pursued in the current study, the high-throughput nature of the organoid production technique might be beneficiary for the clinical translation of ICO for tissue engineering applications. Alternatively, encapsulation of organoids also is of interest for more fundamental research, as more control over organoid growth, size, and cell number can be exerted, and direct co-culture systems can be engineered for probing cell–cell interactions. This can aid in standardizing assays with complex three-dimensional organoid cultures.

### 3.2. Hybrid Microcapsules Produce Uniformly Sized Cholangiocarcinoma Organoids (CCAO)

In order to probe the versatility of the hybrid microcapsules, we also encapsulated CCAO derived from three distinct cholangiocarcinoma patients. We previously initiated these organoids from patient biopsies and confirmed their tumorigenicity [9]. Similar to ICO, the CCAO aggregated to form cellular spheroids, indicating the beneficial environment for cell encapsulation and proliferation of the tumor organoids (Figure 3A). After 21 days of culture, organoids of similar size to the microcapsules were formed, albeit with some patient-specific heterogeneity (Figure 3B). BME-cultured CCAOs utilize their intrinsic capacity to self-organize, which leads to the formation of organoids with high variability in size (Appendix A). However, as stated before, organoid size influences proliferation, in turn potentially influencing drug responses [17]. Single cells dissociated from organoids in BME can be used to perform drug screenings, but these lack the direct cell–cell interactions of organoids. The encapsulated tumor organoids generated herein show a tight regulation of size over time, particularly when compared to gold-standard BME culture (Figure 3C). This demonstrates the efficient generation of standardized tumor organoids using the droplet-generating microfluidic system.

To quantitatively evaluate the potential change in transcriptome due to encapsulation, real-time PCR was employed. KI67, a marker for proliferation, remained constant over time, although with inter-patient heterogeneity, similar to the observations with bright-field microscopy (Figure 3D and Appendix A). Importantly, we found that the stemness of the organoids, as represented by LGR5 and PROM1 expression, was similar in encapsulated CCAO and BME-cultured CCAO (Figure 3E). Thus, the cancer stem cell phenotype, an important characteristic of organoids in BME [41], is not lost during encapsulation. Lastly, integrins play a crucial part in the reciprocal signaling occurring between cells and the extracellular matrix [42]. Integrin signaling is affected by encapsulation within the microcapsules, as ITGB1 is significantly upregulated after 14 and 21 days of encapsulation, suggesting different ECM-cell interactions occur (Figure 3F). Immunofluorescent staining confirmed the cholangiocyte-origin of the tumor organoids with high KRT7 positivity at day 7 and day 21 in all three patient lines (Figure 3G). Furthermore, F-actin was stained to visualize actin bundle alignment, which suggested actin re-arrangement over time (Figure 3H). Overall, encapsulated CCAO exhibit a relatively similar gene and protein expression profile compared to conventional BME culture, and thus, hybrid HLECM and Dex-Ta microcapsules can provide a more standardized, scalable microenvironment.

### 3.3. Encapsulated Cholangiocarcinoma Organoids (CCAO) Are Amendable to Drug Screening

We next investigated the feasibility of the hybrid capsules to be applied to (personalized) drug screens in a controlled format. Gemcitabine and cisplatin combinational chemotherapy is the gold standard for the palliative treatment of CCA patients [43]; thus, we incubated encapsulated CCAO for 72 h with these drugs. After combining 10 µM of cisplatin with different doses of gemcitabine, a clear dose-dependent response on cell viability was detected (Figure 4A). Differences in response were observed between patients, with CCA3 showing the highest sensitivity (IC50 0.1 µM, compared to CCA1 IC50 9.4 µM, CCA2 IC50 2.2 µM). Since dead cells were not able to escape the hybrid microcapsule, drug response could also qualitatively be observed by bright-field microscopy, which showed the accumulation of dead cells (Figure 4B). Encapsulated organoids from CCA1 show a significant difference compared to BME-cultured CCAO after gemcitabine and cisplatin exposure (10 µM: *p* = 0.04, Figure 4C), indicating that encapsulation influences chemo resistance of CCAO in a patient-dependent manner. To expand upon these initial findings, a large-scale drug response experiment was performed to further investigate the potential of encapsulated tumor organoids for throughput personalized medicine drug screening applications. For this, 51 FDA-approved anti-cancer drugs were tested on encapsulated CCAO of the three patients. Patient-derived organoids showed relatively high heterogeneity in drug response to the drug screening panel (Figure 4D). CCA1 had a slightly higher average relative cell viability (0.47 ± 0.41) compared to CCA2 (0.36 ± 0.33) and CCA3 (0.43 ± 0.34). However, no significant differences were present due to the large inter-drug differences. Interestingly, this encapsulated throughput approach identified both drugs that showed resistance in all three patients as well as pan-effective drugs, although the majority of responses showed high patient heterogeneity (Figure 4E). Ceritinib and Romidepsin exhibited the best pan-effective response, with preclinical data in CCA cell lines showing Romidepsin’s ability to induce G2/M phase arrest [44], indicating that future research is warranted. Ceritinib was recently used in a phase I trial for solid tumors, including CCA, with a manageable toxicity profile, paving the way for further development as well [45]. Diving more into potential patient-specific effects, it is clear there is variation in the drug response to singular therapies (Figure 4F). As an example, selumetinib, which previously failed in a phase II study in CCA [42], revealed overall cell viability of 0.01, 0.9, and 1.2 for CCA1, CCA2, and CCA3, respectively. This shows a large difference in response between patients and confirms the observed overall lack of efficacy in the clinical trial. These results stress the need for personalized drug testing and demonstrate the opportunity for high-throughput-produced organoids for scalable personalized medicine. However, probing the similarities between patient and organoid drug response in the future is necessary to validate the ability of standardized production of tumor organoids to uncover pan-effective or patient-specific therapeutics.

## 4. Conclusions

In summary, we established a strategy for the one-step fabrication of hybrid capsules composed of dextran-tyramine and native extracellular matrix using a capillary-based flow focus microfluidic droplet generator that enabled the self-assembly and 3D culture of human cholangiocyte and cholangiocarcinoma organoids. The microcapsules are produced via an easily scalable method that allows for the size-standardized generation of organoids. The established system can facilitate the reduction of size variability conventionally seen in organoid culture by providing uniform scaffolding. As a proof-of-concept, we show the potential of the convergence of organoid technology and microfluidics for drug screening applications by observing patient-specific drug responses. This research offers a scalable platform for fundamental organoid research on cell-ECM interactions as well as translational transplantation and drug discovery applications.

## Figures and Tables

**Figure 1 cells-11-03657-f001:**
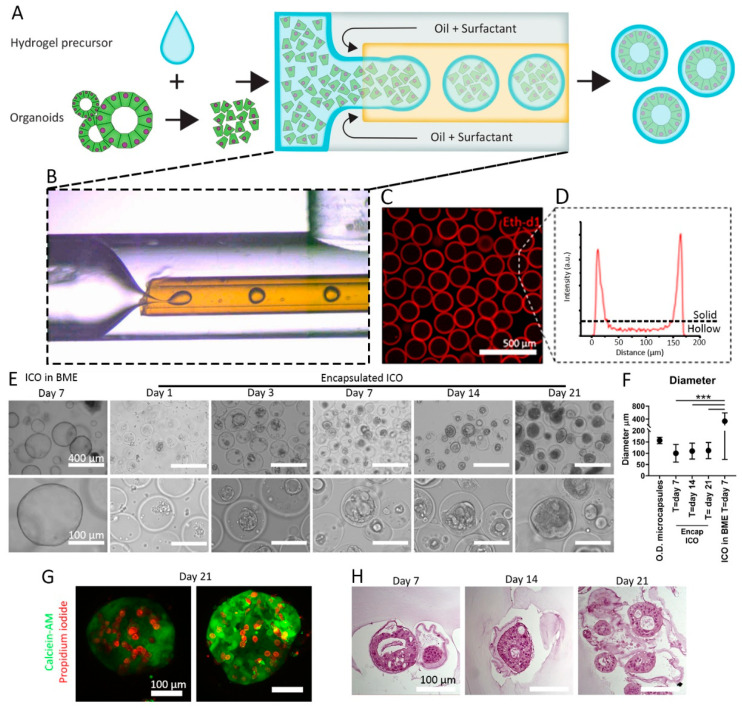
Production of microcapsules containing ICO and liver ECM. (**A**) Schematic of cell encapsulation process with single cells derived from organoids. (**B**) Micrograph of droplet formation in PMMA microfluidic device. (**C**) Confocal fluorescence images of microcapsules of which the Dex-Ta is stained with Ethidium homodimer-1 (Eth-d1). (**D**) Fluorescence intensity histogram of Eth-d1 stained microcapsules. (**E**) Bright-field microscope images of cholangiocytes cultured in BME control for 7 days or cultured in microcapsules for 1, 3, 7, 14, and 21 days. (**F**) Average diameter of microcapsules (O.D. = outer diameter) encapsulated ICO at days 7, 14, and 21, and control ICO grown in BME at day 7. The graph indicates mean diameter + standard deviation. Significance was tested using ANOVA. *** *p* < 0.001. (**G**) Representative fluorescence images of ICO in microcapsules stained with calcein-AM in green for live cells and propidium iodide in red for dead cells. (**H**) Bright-field microscope images of hematoxylin and eosin-stained ICO cultured for 7, 14, and 21 days.

**Figure 2 cells-11-03657-f002:**
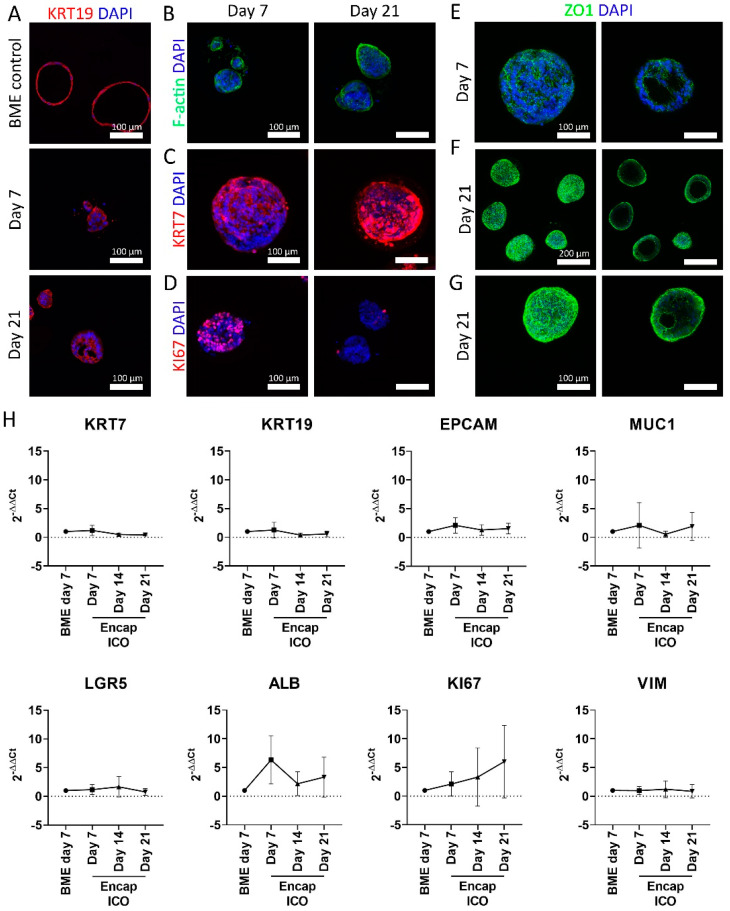
Assessment of cholangiocyte markers of ICO cultured in microcapsules. (**A**) Confocal fluorescence images of BME control ICO cultured for 7 days and microencapsulated ICO cultured for 7 and 21 days stained for KRT19 in red and DAPI in blue. (**B**) Confocal fluorescence images of encapsulated ICO cultured for 7 and 21 days, stained for F-actin in green and DAPI in blue, (**C**) KRT7 in red and DAPI in blue, (**D**) KI67 in red and DAPI in blue, (**E**–**G**) ZO1 in green and DAPI in blue. The left column represents the max projection of the entire Z-stack, whereas the right column only shows the max projection of a selection of the Z-stack so that the lumen of the ICO is visible. (**H**) Gene expression of BME control and encapsulated ICO cultured for 7, 14, and 21 days of KRT7, KRT19, EPCAM, MUC1, LGR5, ALB, KI67, and VIM (*n* = 5). No statistically significant differences were observed. Scale bars for (**A**–**E**) and (**G**) indicate 100 µm. Scale bar for (**F**) indicates 200 µm.

**Figure 3 cells-11-03657-f003:**
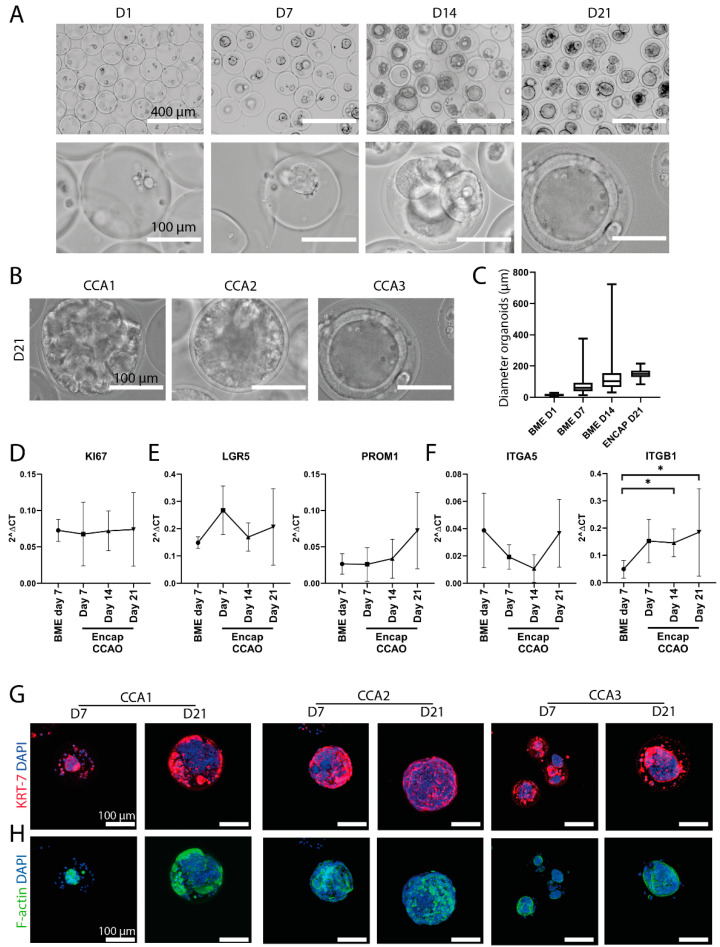
CCAO formation and characterization in microcapsules. (**A**) Representative bright-field microscope images of cholangiocarcinoma tumor cells cultured in microcapsules for 1, 7, 14, and 21 days. Scale bar indicates 400 µm in the top row and 100 µm in the bottom row. (**B**) Representative bright-field micrographs of three distinct patient-derived CCAO after 21 days of culture in microcapsules. Scale bar indicates 100 µm. (**C**) Size analysis of CCAO in BME or encapsulated in microcapsules. Shown is the mean and standard deviation of individual organoids grown in BME (Day 1: *n* = 46, Day 7: *n* = 73, Day 14: *n* = 73) or microcapsules (*n* = 117). Standard deviation was smallest in microcapsules. (**D**–**F**) Gene expression analysis using real-time PCR of BME control and encapsulated CCAO cultured for 7, 14, and 21 days of KI67 (**D**), LGR5, PROM1 (**E**), and ITGA5 and ITGB1 (**F**). * *p* < 0.05 (**G**,**H**) Confocal fluorescence imaging of encapsulated CCAO cultured for 7 or 21 days, which stained for KRT7 in red (**G**) and F-actin in green (**H**). Both images also contain DAPI staining in blue. Scale bar indicates 100 µm.

**Figure 4 cells-11-03657-f004:**
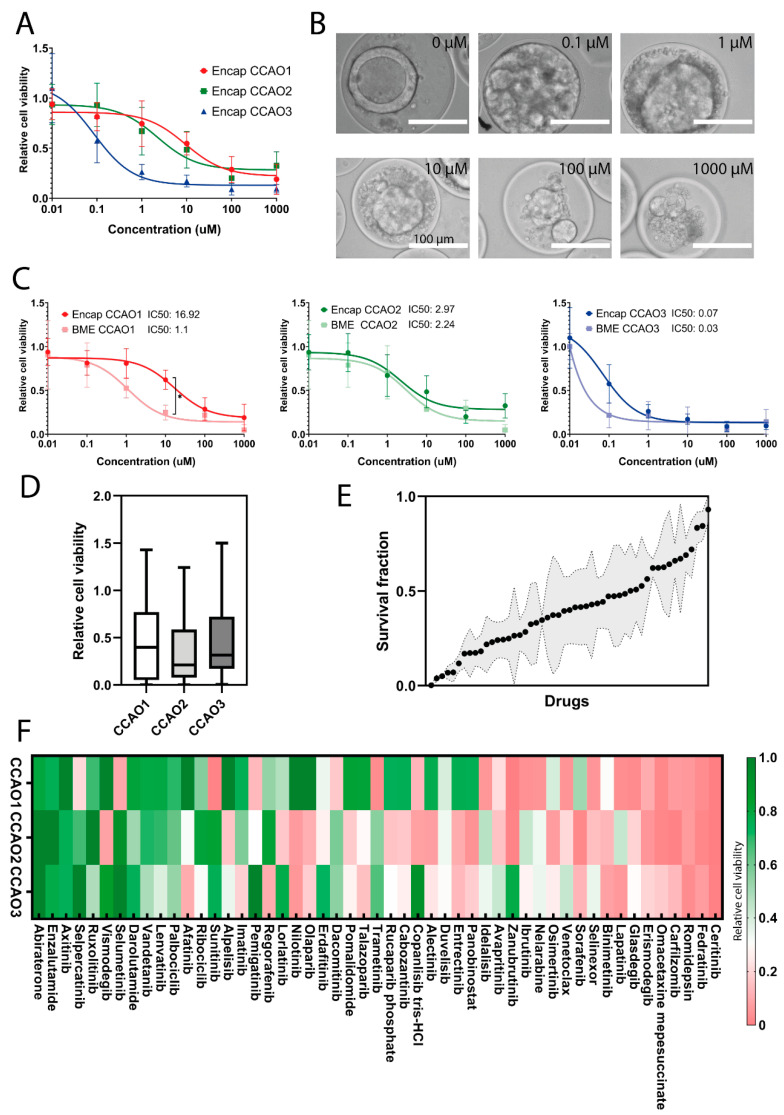
Cell response of encapsulated CCAO in drug screening applications. (**A**) Quantification of relative cell viability in encapsulated CCAO of 3 distinct donors after exposure to a fixed concentration of cisplatin and variable concentrations of gemcitabine after 72 h. Error bars indicate standard error of mean. Nonlinear regression curves were fitted to the data to determine IC50 values. (**B**) Representative bright-field micrographs of encapsulated CCAO1 organoids after exposure to 0 µM, 0.1 µM, 1 µM, 10 µM, 100 µM, and 1000 µM gemcitabine and 10 µM cisplatin. Scale bar indicates 100 µm. (**C**) Comparison of relative cell viability for encapsulated CCAO and BME-cultured CCAO after exposure to a fixed concentration of cisplatin and variable concentrations of gemcitabine after 72 h. Error bars indicate standard error of mean. Nonlinear regression curves were fitted to the data to determine IC50 values. * indicates a statistically significant difference in drug response between CCAO1 in BME and encapsulation. (**D**) Quantification of average cell viability upon exposure to 51 different chemotherapeutic drugs. (**E**) Box plot displaying the average survival fraction of CCAO ranked from most effective to the least effective drug. Black dot indicates the average survival fraction, and the gray area represents the standard deviation. (**F**) Cluster map of the three different patient-derived CCAO responses to the drug response panel. Color corresponds to the relative cell viability: green represents resistance to the drugs, and red represents a sensitive response.

## Data Availability

Data available on request.

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
