# Peer review of "Scalable Production of Size-Controlled Cholangiocyte and Cholangiocarcinoma Organoids within Liver Extracellular Matrix-Containing Microcapsules"

_cells, 2022, doi:10.3390/cells11223657_

Round 1

Reviewer 1 Report

The study aims to produce "size-controlled" cholangiocytes or cholangioca organoids in microcapsules. 

Comments

1. The size-controlled organoid is actually the microcapsule produced by microfluidic glass capillary system, not the organoid itself.  Did the authors mix the term of the microcapsule as organoid? 

2. How many cells in each organoid in the study? What percentage of space did the organoid(s) occupy in a microcapsule?

3. Can 3D culture of cholangiocarcinoma be called organoid? Did these cancer formulate any "organoid" structure (like Figure 1H)?   

Reviewer 2 Report

This paper is interesting and valuable for researchers on biomaterials, cancer tissue engineering, or device. However, the authors should discuss the study by comparing related recent papers, especially biomaterial-usage 3D cancer models. In the current manuscript, it is impossible to understand the novelty or strength of this study. The authors should introduce the represented biomaterial (especially spherical or encapsulated) for 3D models. Taken together, major revision should be made before re-submission. The manuscript would be re-considered only when all the comments are responded.

1. Introduction or discussion

The authors should introduce the represent biomaterial for 3D models. In addition, the strength of this study should be indicated by comparing the research. Their biomaterials can assist the ECM component and are important for the authors’ research, so the authors should clarify the difference between them. The reviewer recommends that the paper to be quoted.

Overview (for concept)

Cancers 202012(10), 2754

Alginate

https://doi.org/10.1016/j.biomaterials.2015.11.030

Collagen

https://doi.org/10.1073/pnas.1212834109

Gelatin

Tissue Eng. Part C Methods 201925, 711–720. https://doi.org/10.1089/ten.tec.2019.0189

Matrigel

Scientific Reports volume 8, Article number: 5333 (2018

2. Discussion

This part can not be understood. The authors should discuss the strength by quoting related papers above and comparing these researches.

Reviewer 3 Report

Dear Authors,

This is a very interesting paper which shows a novel method to improve organoid reproducibility and standardisation through micro encapsulation of organoids in liver ECM. This paper goes on to describe how it could be used for patient derived drug discovery and hence personalised medicine with significant benefits.

I can see that there is a significant amount of work done here, although the results are limited by the small number of patient samples and high heterogeneity. It would have been ideal to have comparisons with simple patient derived organoids, not using this technique, or a comparison using standardised CCA cell line derived organoids with an established phenotype and known chemotherapy sensitivities.

I have no overall criticisms of the paper as it is a proof-of-concept which is demonstrated well here.

Author Response

We would like to thank the reviewer for reading our manuscript and responding to it. We understand that the heterogeneity between donors or patients can make interpretation of data more complicated. However, we also feel that there is an added benefit of having the ability to work with human patient material. Nevertheless, we agree future work should focus on benchmarking strategies to other available models, including potentially CCA cell lines and ‘simple’ organoids. 

Reviewer 4 Report

The manuscript in front of me is dealing with scalable production of size-controlled cholangiocyte and 2 cholangiocarcinoma organoids within liver extracellular 3 matrix-containing microcapsules. The manuscript is well written and well organized. The list of references should be improved. I am not sure about ref. 18, since the publication year is 2023. Is it ok?

Author Response

We would like to thank the reviewer for reading our manuscript and providing feedback. We understand the confusion about reference 18, as it says publication year is 2023. This is because the referred manuscript is accepted and published early on-line, while it will only be featured in the physical print issue of Bioactive Materials of January 2023 (doi: https://doi.org/10.1016/j.bioactmat.2022.04.005). For this reason, we have already included it in our manuscript. 

Round 2

Reviewer 1 Report

.

Reviewer 2 Report

I recommend the publication.